# Training-free Prompt Placement by Propagation for SAM Predictions in 3D Bone CT Scans

**Caroline Magg**[1,2]                              C.MAGG@AMSTERDAMUMC.NL

**Lukas P.E. Verweij**[2]

**Maaike A. ter Wee**[2]

**George S. Buijs**[2]

**Johannes G.G. Dobbe**[2]

**Geert J. Streekstra**[2]

**Leendert Blankevoort**[2]

**Clara I. Sánchez**[1,2]

[1] *University of Amsterdam, Amsterdam, The Netherlands*

[2] *Amsterdam UMC location University of Amsterdam, Amsterdam, The Netherlands*

**Editors:** Accepted for publication at MIDL 2024

## Abstract

The Segment Anything Model (SAM) is an interactive foundation segmentation model, showing impressive results for 2D natural images using prompts such as points and boxes. Transferring these results to medical image segmentation is challenging due to the 3D nature of medical images and the high demand of manual interaction. As a 2D architecture, SAM is applied slice-per-slice to a 3D medical scan. This hinders the application of SAM for volumetric medical scans since at least one prompt per class for each single slice is needed. In our work, the applicability is improve by reducing the number of necessary user-generated prompts. We introduce and evaluate multiple training-free strategies to automatically place box prompts in bone CT volumes, given only one initial box prompt per class. The average performance of our methods ranges from 54.22% Dice to 88.26% Dice. At the same time, the number of annotated pixels is reduced significantly from a few millions to two pixels per class. These promising results underline the potential of foundation models in medical image segmentation, paving the way for annotation-efficient, general approaches. The code is available at this github URL.

**Keywords:** SAM, foundation segmentation model, bone segmentation, CT scans

## 1. Introduction

Medical image segmentation (MIS) plays a crucial role in various clinical applications, among others the segmentation of bones as part of the musculoskeletal system for medical interventions and for examining changes in anatomical structures (Bonaldi et al., 2023). In order to reduce user interaction and time-consuming manual delineations, deep learning (DL) methods have been developed to automatically generate a segmentation mask of diverse bone structures in CT imaging (Lindgren Belal et al., 2019; Deng et al., 2022; Wasserthal et al., 2023; Magg et al., 2024). However, the best performing methods so far developed are trained in a fully supervised manner requiring pixel-based annotation. These models cannot be transferred to other unseen structures without new annotations and training. Foundation Models (FMs) offer better generalisability as they are general-purpose models that are trained on large data sets and can be used for a wide range of

downstream tasks. New advances for segmentation FMs have been made with the prompt-based Segment Anything Model (SAM) by Kirillov et al. (2023). This model relies on the user to mark a point in the object or bound the region of interest, which is referred to as placing a prompt. However, current segmentation FMs are not suitable for MIS because of two main challenges. First, SAM has been trained on natural images. Several studies (He et al., 2023; Mazurowski et al., 2023; Ma et al., 2023; Cheng et al., 2023a) evaluated SAM for MIS and reported unstable performance across diverse datasets which can be explained by the difference between natural and medical images. To overcome this challenge, adaptation techniques (Wu et al., 2023; Li et al., 2023) and fine-tuning on medical data (Ma et al., 2023; Cheng et al., 2023b; Xie et al., 2024) are introduced. Second, the 2D SAM architecture does not account for the 3D nature of volumetric medical scans. As at least one prompt per slice is required, the interactive approach depends on a high manual interaction if applied to volumetric MIS. This reduces the applicability in high-throughput or less supervised medical tasks. To alleviate the second challenge, we propose multiple training-free strategies that automatically place box prompts for bone CT scans. In contrast to our training-free method, Lei et al. (2023) and Wang et al. (2023) addressed this challenge using approaches that required an extra training phase, either to predict bounding boxes locations or to adapt SAM architecture to 3D, respectively. Most similar to our work is EviPrompt by Xu et al. (2023) which is an evidence-based method for generating point prompts for SAM based on uncertainty estimates. A medical image-annotation pair is used as reference to create a shared embedding in the feature space and extract pixel similarity compared to the sample of interest.

In this work, we propose four different training-free strategies of prompt placement by propagation to adapt SAM for bone segmentation in 3D CT scans. In contrast to EviPrompt, only one initial box prompt is required and the remaining box prompts are inferred from predictions in adjacent slices. We evaluated our methods on three datasets containing different bone structures, and showed the annotation-efficiency of our methods by analyzing performance with respect to the required numbers of annotated pixels.

## 2. Data

Three different datasets are used for our experiments, i.e., two internal datasets, namely Ds and Dk, and one publicly available dataset, namely Dt (see Appendix A, Figure 4). The first internal dataset (Ds) contains 15 bilateral CT scans of the shoulder joint with annotations of the left and right humerus and scapula. The shoulder CT scans were acquired with a Brilliance 64-channel CT Scanner (Philips Healthcare, Best, The Netherlands) with 250 mAs, 120 kV, pixel resolution of $0.83 - 0.98$mm and slice thickness $0.3 - 1.0$ mm. The second internal dataset (Dk) consists of 25 unilateral knee CT scans with knee implants, including 20 cadaveric and 5 patient scans. The tibial cortex bone and the tibial component of a knee prosthesis are labeled in the knee CT scans which were acquired with a Brilliance 64-channel CT Scanner (Philips Healthcare, Best, The Netherlands) or a Siemens SOMATOM Force with 160 mAs, 120 kV, isotropic voxel spacing of 0.45mm. The internal datasets were annotated using an in-house annotation software (Dobbe et al., 2014) and ITK-Snap (Yushkevich et al., 2006). The external dataset, Dt, is extracted from a publicly available dataset (https://doi.org/10.5281/zenodo.6802613, Wasserthal et al. (2023)). Two different

subsets, Dt1 and Dt2, are extracted from the provided test set: Dt1 consists of 33 CT scans containing the left and right humerus and scapula, in analogue to Ds; and Dt2 consists of 53 CT scans containing the left and right hip and femur. As the smallest dataset, Ds was used for the method development and preliminary hyperparameter testing. Dk and Dt were only used for additional evaluation purposes.

## 3. Methods

Based on the prompt-based mode of SAM, our method is an automated prompt placement approach which is initialized by defining a box prompt. Assuming that the boundary of the object of interest in a 3D CT scan, in our case bone structures, can be modeled by a smooth surface over slices, local changes of prompts between neighboring slices could be considered minimally. Based on this, we present four different strategies that iterative propagate predictions from adjacent slices to infer a new prompt.

### 3.1. Revisiting SAM

SAM by Kirillov et al. (2023) is a prompt-based FM developed for interactive 2D segmentation. The architecture consists of three components: an image encoder, a prompt encoder and a mask decoder. First, the input image $I \in \mathbb{R}^{H \times W}$ is embedded by a Vision Transformer (ViT) pre-trained as Masked Auto-encoder (MAE). Then, the prompt encoder creates a prompt encoding from geometric prompts (namely bounding box or point). Multiple studies illustrate that box prompts achieve better results compared with point prompts (Ma et al., 2023; Mazurowski et al., 2023; He et al., 2023). Thus, we focus on box prompts $P \in \mathbb{N}^{2 \times 2}$ represented by an embedding pair, i.e., the positional encoding of its top-left and bottom-right corner. Finally, a lightweight mask decoder maps the image and prompt embedding to an output mask $M = f_{SAM}(I, P)$ with $M \in \mathbb{R}^{H \times W}$.

### 3.2. Prompt Inference by Propagation

For volumetric data, SAM is currently applied in a slice-by-slice manner per class (Mazurowski et al., 2023). Therefore, given a 3D CT scan $I \in \mathbb{R}^{H \times W \times D}$ with the i-th slice $I_i \in \mathbb{R}^{H \times W}$ and the corresponding mask prediction $M \in \mathbb{R}^{H \times W \times D}$ with $M_i \in \mathbb{R}^{H \times W}$, one prompt $P_i$ for each slice $i$ is required, so that $M_i = f_{SAM}(I_i, P_i)$. In our approaches, $P_i$ is defined as a bounding box enclosing the object of interest and described by $P_i = f_{BOX}(M_i) = \{(x_i^t, y_i^t), (x_i^b, y_i^b)\}$, with the coordinates of the top-left $(x_i^t, y_i^t)$ and bottom-right $(x_i^b, y_i^b)$ pixel. Given an initial slice $S_j \in I$ and a manually provided prompt $P_j$, we propose four approaches to estimate $P_i$, and thus obtain $M_i$. Figure 1 visualizes the idea behind three of the strategies. Without any loss of generality, $j < i$ applies in the following.

**Baseline** ($f_b$): In this approach, given the initial slice, the box prompt is propagated uni-directional through the volume, inferred from the previous prediction as follows:

$$M_i = f_b(I_i, M_{i-1}) = f_{SAM}(I_i, P_i), \text{ with } P_i = f_{BOX}(M_{i-1}). \tag{1}$$

**Stochastic Approach** ($f_s$): For this approach, the box prompt is also propagated in one direction, i.e., uni-directional, but relies on multiple modifications of the previous

prompt prediction. Specifically, $K$ prompts are generated by randomly shifting the coordinates of the box enclosing the previous prediction as follows:

$$P_i^{(k)} = \{(x_i^t, y_i^t) + (\delta_x^{(k)}, \delta_y^{(k)}), (x_i^b, y_i^b) + (\delta_x^{(k)}, \delta_y^{(k)})\}, \forall k = \{1, \ldots, K\}, \text{ with } \delta_x^{(k)}, \delta_y^{(k)} \in \mathbb{N}.$$

Based on this, $M_i$ is estimated as follows:

$$M_i = f_s(I_i, M_{i-1}, K) = \bigcap_{k=1}^{K} f_{SAM}(I_i, P_i^{(k)}) \tag{2}$$

**Nested Approach ($f_n$):** The nested approach performs bi-directional box prompt propagation. The information of the previous prediction $M_{i-1}$ and the following slice $I_{i+1}$ are used to perform propagation steps forwards and backwards as follows:

$$M_i = f_n(I_i, I_{i+1}, M_{i-1}) = f_{SAM}(I_i, P_i), \text{ with}$$
$$P_i = f_{BOX}(f_{SAM}(I_{i+1}, f_{BOX}(f_{SAM}(I_i, f_{BOX}(M_{i-1}))))). \tag{3}$$

**Combined approach ($f_c$):** The nested and stochastic approaches can be combined as

$$M_i = f_c(I_i, I_{i+1}, M_{i-1}, K) = f_s(I_i, f_s(I_{i+1}, f_s(I_i, M_{i-1}, K), K), K). \tag{4}$$

by using $f_s$ from Equation (2).

The initial slice $S_j$ is chosen randomly around the center of the object, because there will be a higher certainty of the correctness of the manually placed prompt $P_j$ due to the larger object area contained in this region. The obtained masks are postprocessed in two steps. First, the largest component associated with the prediction generated by the initial prompt is retained and all other disconnected components are removed (pp1). Second, predictions are removed if the corresponding box is significantly smaller than the initial box (pp2).

## 4. Experimental Setup

For our experiments, we used the pre-trained SAM ViT-B, as described in (Kirillov et al., 2023). The 2D architecture processes the 3D CT scans in axial slices, since the axial plane has usually a higher resolution in CT scans. For each slice, a binary mask per class is generated. In order to combine all binary masks to one multi-class mask, a class processing sequence was defined: For Dk and Dt1, humerus before scapula; for Dk, tibia bone before tibial implant; for Dt2, hip before femur. Following Mazurowski et al. (2023), a maximum of three bounding boxes, each larger than 5% of the 2D mask area, was extracted from a prediction. The values $\delta_x^{(k)}$ and $\delta_y^{(k)}$ are sampled from a uniform distribution between 0 and 5. The size of threshold in pp2 is empirically set to 10% of the size of $P_j$. Other parameters, such as the initial slice position and $K$, were studied in hyperparameter experiments on Ds without postprocessing and after pp1 to determine optimal values.

As comparison to FMs, two 3D full resolution nnUNets (Isensee et al., 2021) for Ds and Dk were trained. The training details are reported in Appendix B. For Dt, the trained nnUNet from Wasserthal et al. (2023) was deployed. Another 3D full resolution nnUNet was trained

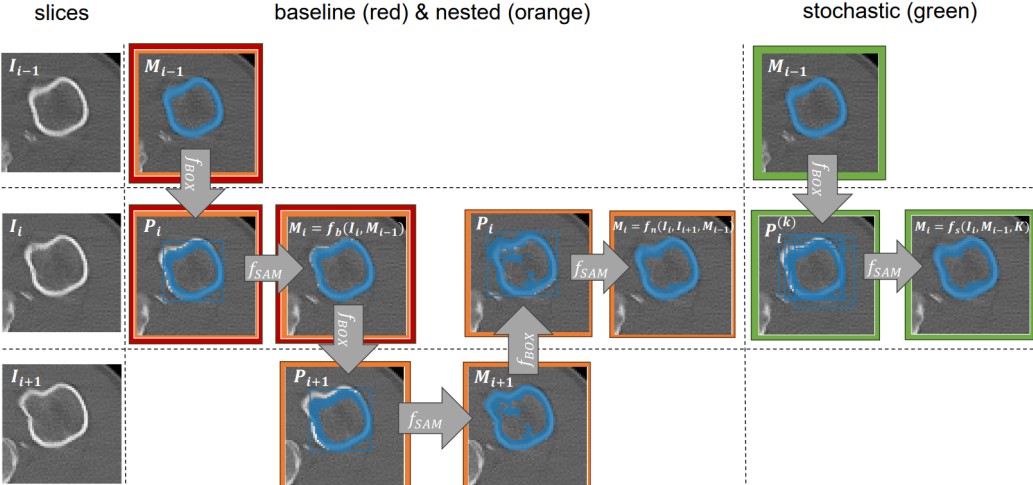

Figure 1: Overview of the three strategies for prompt placement in slice $I_i$. The boxes are color-coded based on the strategy: baseline (red), nested (orange) and stochastic (green). For the sake of simplicity, the combined approach is omitted.

on predictions from our best method as reference labels, using the first fold of Ds and the settings mentioned in Appendix B. This demonstrates the capability of our methods to create pixel-annotations for a fully supervised training, starting with one bounding box per object. As an upper performance estimate for our methods, SAM was evaluated based on the evaluation scheme of Mazurowski et al. (2023), where a maximum of three box prompts were extracted from the ground truth labels. This mimics the placement of the most ideal SAM box prompts when manually entering one prompt per slice.

All obtained masks were compared with the ground truth labels by means of two common segmentation metrics, i.e. Dice Coefficient (Dice) and the Hausdorff Distance 95% (HD95). In addition, we analysed the number of annotated pixels in relation to the Dice to investigate the annotation-efficiency.

## 5. Results

The evaluation results of our prompt propagation approaches without postprocessing (no pp) and after applying both postprocessing steps (pp), the traditional SAM, and the nnUNet predictions are shown in Table 1. The methods are not statistically significantly different (see Appendix C.1). Results for only applying one of both postprocessing steps are shown in Appendix C.2, Table 3. For Ds and Dt1, qualitative examples are shown in Figure 2. Figure 3 shows the annotation-efficiency of our approaches compared to fully supervised methods. An ablation study about the number of prompts used for initialization considering performance and annotation time is given in Appendix C.3. In Appendix C.4, performance across different slice ranges surrounding the initial slice is evaluated to understand performance evolution across volumes. The results after training a nnUNet with the masks obtained by combination pp1 are 90.19% Dice and 2.77mm HD95, while for training a nnUNet with the ground truth labels the results are 98.35% Dice and 0.92mm HD95.

According to Figure 8 in Appendix C.5, the best results in the hyperparameter experiments are achieved with the following settings: The initial slice is sampled from $-25\%$ to $+25\%$ around the object's center. For the stochastic-based approach, $K$ is set to 10 and if at least 5 predictions include a particular pixel, it is included in the final mask.

Table 1: Results of our methods, the reference SAM, and nnUNet. For our methods, the results without postprocessing (no pp) and after postprocessing (pp) are reported. For Dt, the nnUNet results marked with * are taken from Wasserthal et al. (2023).

| Method | Dice (%) ↑ | | HD95 (mm) ↓ | | Dice (%) ↑ | | HD95 (mm) ↓ | |
|---|---|---|---|---|---|---|---|---|
| | *Ds* | | | | Dt1 | | | |
| nnUNet | 98.5 | | 0.9 | | 91.5* | | not reported* | |
| SAM | 90.0 | | 2.5 | | 84.2 | | 5.9 | |
| | *no pp* | *pp* | *no pp* | *pp* | *no pp* | *pp* | *no pp* | *pp* |
| baseline | 84.4 | 87.6 | 22.3 | 4.2 | 77.2 | 78.5 | 31.8 | 15.2 |
| stochastic | 85.5 | 87.6 | 17.0 | 4.0 | 70.3 | 69.6 | 23.9 | 25.5 |
| nested | 85.1 | 88.1 | 21.6 | 4.1 | 78.0 | 76.8 | 24.5 | 16.8 |
| combined | 85.8 | 87.9 | 18.6 | 4.5 | 70.3 | 69.1 | 25.3 | 26.3 |
| | Dk | | | | Dt2 | | | |
| nnUNet | 95.6 | | 0.6 | | 95.1* | | not reported* | |
| SAM | 76.3 | | 3.5 | | 92.0 | | 5.4 | |
| | *no pp* | *pp* | *no pp* | *pp* | *no pp* | *pp* | *no pp* | *pp* |
| baseline | 54.2 | 54.8 | 27.1 | 24.3 | 78.2 | 77.7 | 32.6 | 29.9 |
| stochastic | 58.1 | 58.1 | 20.1 | 19.7 | 80.8 | 80.7 | 18.8 | 19.5 |
| nested | 55.7 | 56.8 | 31.1 | 23.4 | 79.5 | 78.6 | 30.8 | 28.9 |
| combined | 58.1 | 58.9 | 21.2 | 19.4 | 80.6 | 80.0 | 18.9 | 18.8 |

## 6. Discussion

In this work, we have presented multiple training-free strategies to apply SAM with box prompts to 3D bone CT scans starting with only two pixels annotated. We have introduced four methods, following different strategies of using the adjacent information. Our experiments do not reveal a definitive trend favoring one of our methods. However, our methods significantly reduce the number of annotated pixels and compared to using SAM in the traditional slice-by-slice manner or a fully supervised segmentation method, while maintaining a certain level of segmentation performance and showing promising results.

Metal artefacts, similar structures in close proximity to each other or interlocking structures, as well as objects that split into multiple components have a negative effect on our training-free strategies. In some cases, they can lead to segmentation overflow to surrounding structures or a lack of components tracking during a split (see Figure 12). Aside from that, a lower contrast between bone and background decreases the performance of our methods, as seen in Dt1, aligning with the limitations of SAM (Mazurowski et al., 2023). However, our proposed strategies show very good results when the segmentation task is easier and there is a good bone-background contrast. For example, for Ds, the Dice Coefficients are only $2-3\%$ off the traditional SAM (see Table 1). As a very thin structure, the

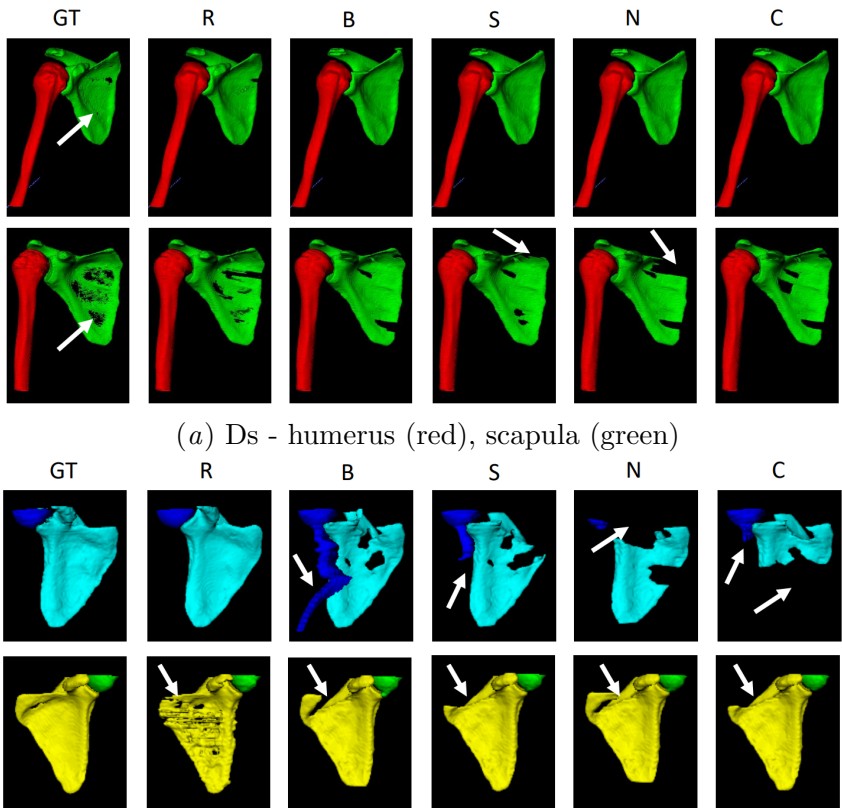

(a) Ds - humerus (red), scapula (green)

(b) Dt1 - humerus (dark blue, green), scapula (bright blue, yellow)

Figure 2: Examples of 3D models for Ds (a) and Dt1 (b).

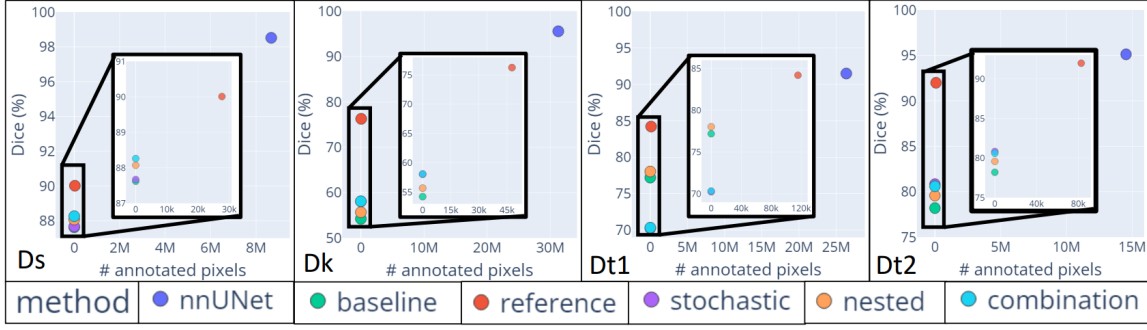

Figure 3: Analysing performance (Dice) wrt number of annotated pixels.

scapula is difficult to manually annotate in a smooth 3D manner, resulting in some holes in the ground truth due to annotation errors (see Figure 4 Ds). Even when the reference standard is not completely correct, our methods produce smooth surfaces (see Figure 2(a)).

Analysing the individual methods reveals that baseline and nested approach can show severe over-propagation if an error is propagated (see Figure 12). In contrast, stochastic-based methods are more conservative, requiring agreement between multiple predictions. However, they suffer from vanishing boxes and thus, missing segmentation, when agreement

is lacking (see Figure 2(*b*), Figure 12(*b*)). Combining stochastic and nested strategy does not balance weaknesses and may even amplify them (see Figure 12). Our current postprocessing steps target two reoccurring segmentation mistakes (see Figure 11). SAM can predict multiple disconnected components for one box, detecting structures similar to the object of interest. Removing all components, that are not connected to the largest component predicted from the initial box (pp1), improves performance. Moreover, SAM lacks spatial context, leading to the box not always vanishing at the top and bottom of the object. Restricting the box size (pp2), corrects for small additional structures. Using only pp1 yields greater performance gains, as the corrections are generally larger than pp2 corrections made at the object's boundary (see Table 3, Figure 11). However, since each postprocessing step addresses another error pattern, the best results are achieved applying both (see Table 1).

As shown in Figure 3, there is a trade-off between performance and annotated pixels. While nnUNet consistently outperforms other solutions, it requires extensive ground truth labels for each new task and shift in data characteristics. In contrast, traditional SAM reduces annotated pixels by a magnitude of 2, and it seems robust to data and task changes. However, SAM lacks scalability for high-throughput 3D segmentation tasks due to the demand of manual interactions per slice. Our approaches are independent of the number of slices. Figure 5 shows that more user-provided prompts enhance performance in our approaches until reaching the fully prompted SAM level, which comes with the cost of additional annotation time. The annotation-efficient predictions can be used for pseudo-label training or as a starting point for model-assisted labeling. Training nnUNet with SAM predictions (see Figure 13) achieves 90% Dice, demonstrating the feasibility of creating a training dataset for supervised learning. Aside annotated datasets, our study provides insights for selecting training samples for efficient fine-tuning. For example, analyzing performance evolution throughout the volume reveals that for some slices the performance is closer to nnUNet and better than fully-prompted SAM (see C.4). Thus, our methods have value for some slices, but more challenging ones may require supervised approaches.

A limitation of our method is relying solely on prediction information from adjacent slices, excluding image information. To enhance segmentation performance without increasing annotation effort, incorporating a priori information like shape could be beneficial. Further improvements in prediction quality could be achieved by exploiting the full range of geometric prompts, combining box with positive and negative point prompts to reduce ambiguity. The performance of our methods depends on the underlying SAM version, which should be investigated further in combination with models fine-tuned on medical data, like Med-SAM (Wu et al., 2023) or SAM-Med2D (Cheng et al., 2023b). It is important to note that our goal is not to achieve state-of-the-art results but to offer a training-free strategy for using prompt-based 2D segmentation FMs, like SAM, in 3D medical data with one prompt per class per volume rather than one prompt per slice.

## 7. Conclusion

We have presented multiple training-free strategies to apply the box-prompt mode of SAM to 3D bone CT scans with only one initial box prompt. Our method significantly decreases the number of annotated pixels while maintaining a certain level of segmentation performance. This work is another step towards the applicability of FMs for 3D bone segmentation.

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

## Appendix A. Data Overview

Figure 4 gives an overview of the three datasets with an example of a CT slice and 3D model of the reference masks.

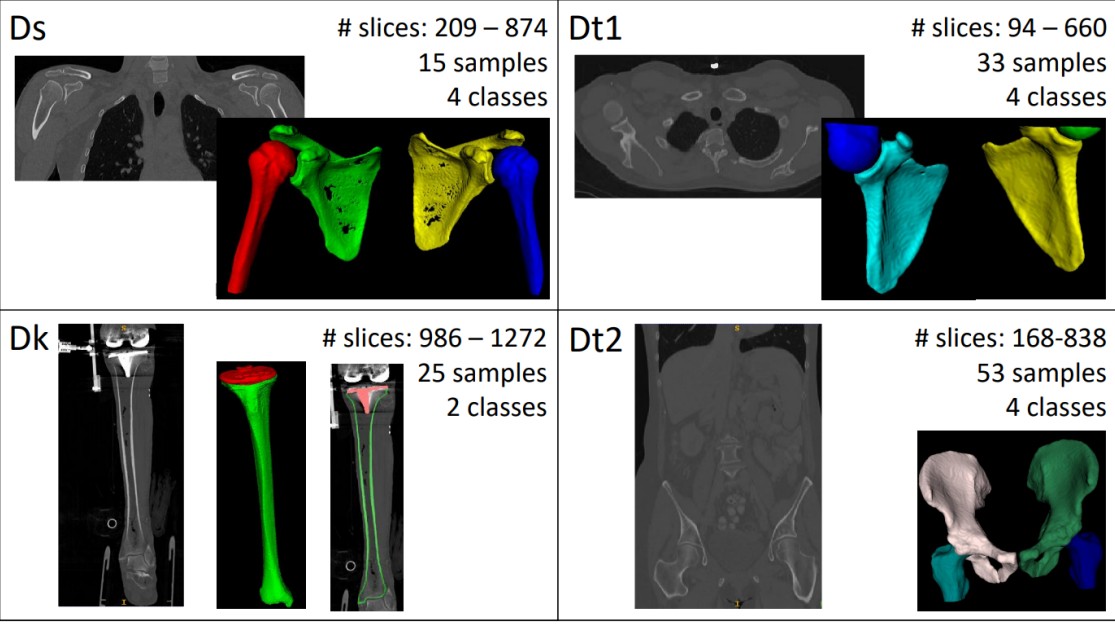

Figure 4: Dataset overview with exemplary CT slices and 3D models of the reference mask.

## Appendix B. nnUNet Training Details

We train two 3D full resolution nnUNet (Isensee et al., 2021) on each of the datasets Ds and Dk. The default training settings have been retained, except for mirroring in data augmentation. For Ds, the mirroring on the vertical axes would confuse the left and right labels in the bilateral scans and is therefore removed. For Dk, the mirroring on the horizontal axes is removed since a horizontally flipped femoral implant component shows some similarity with a tibial implant component. The models are trained and evaluated on a 5-fold and 4-fold patient-based cross-validation split for Ds and Dk, respectively. The training is performed on an NVIDIA Geforce RTX 2080 Ti 12GB and an Intel Core Xeon Gold 6128 3.40GHz CPU.

## Appendix C. Ablation studies

Ablation studies for statistical significance, postprocessing, the number of prompts used for initialization, and hyperparameter settings have been performed.

### C.1. Statistical significance

For each dataset, the statistical significance is tested with bootstrapped paired t-tests using 1000 iterations. Due to the high number of tests per dataset, Bonferroni correction is used, i.e., corrected $\alpha = 0.05/6 = 0.0083$. The results for each method combination per dataset is shown in Table 2. There is no a significant difference in the methods.

Table 2: P-values for statistical significance testing of difference in methods per dataset. No pairing shows statistical significance, compared to the corrected $\alpha = 0.0083$.

| Method combination | Ds | Dk | Dt1 | Dt2 |
|---|---|---|---|---|
| baseline, stochastic | 0.918 | 0.068 | 0.052 | 0.372 |
| baseline, nested | 0.027 | 0.029 | 0.102 | 0.359 |
| baseline, combined | 0.276 | 0.039 | 0.009 | 0.433 |
| stochastic, nested | 0.046 | 0.295 | 0.111 | 0.382 |
| stochastic, combined | 0.44 | 0.292 | 0.760 | 0.625 |
| nested, combined | 0.445 | 0.093 | 0.058 | 0.544 |

### C.2. Postprocessing

Table 3 shows the results for our methods without postprocessing (no pp) and after applying only one postprocessing step, i.e., either pp1 or pp2.

Table 3: Results of our methods without postprocessing (no pp), and after one postprocessing step, either only pp1 or only pp2.

| Method | Dice (%) ↑ | HD95 (mm) ↓ | Dice (%) ↑ | HD95 (mm) ↓ |
|---|---|---|---|---|
| | *Ds* | | *Dt1* | |
| | *no pp, pp1, pp2* | *no pp, pp1, pp2* | *no pp, pp1,pp2* | *no pp, pp1,pp2* |
| baseline | 84.4, 86.4, 85.6 | 22.3, 6.6, 13.2 | 77.2, 77.5, 0.8 | 31.8, 30.8, 15.5 |
| stochastic | 85.5, 87.7, 85.6 | 17.0, 4.0, 15.6 | 70.3, 70.2, 70.2 | 23.9, 24.5, 25.0 |
| nested | 85.1, 88.0, 85.7 | 21.6, 4.3, 14.1 | 78.0, 77.8, 77.1 | 24.5, 21.1, 18.3 |
| combined | 85.8, 88.3, 85.7 | 18.6, 3.9, 14.6 | 70.3, 70.3, 69.1 | 25.3, 25.9, 25.7 |
| | *Dk* | | *Dt2* | |
| | *no pp,pp1, pp2* | *no pp,pp1,pp2* | *no pp, pp1,pp2* | *no pp, pp1,pp2* |
| baseline | 54.2, 54.4, 54.7 | 27.1, 26.9, 24.4 | 78.2, 77.3, 78.6 | 32.6, 32.9, 29.4 |
| stochastic | 58.1, 58.2, 57.9 | 20.1, 19.9, 20.01 | 80.8, 80.8, 80.8 | 18.8, 19.0, 19.3 |
| nested | 55.7, 56.0, 56.5 | 31.1, 30.2, 24.4 | 79.5, 78.3, 79.8 | 30.8, 30.7, 28.8 |
| combined | 58.1, 58.4, 58.6 | 21.2, 20.6, 19.9 | 80.6, 80.1, 80.4 | 18.9, 18.9, 19.5 |

### C.3. Number of prompts & annotation time

In order to evaluate how the performance changes if multiple prompts are used for initialization, the number of initialized slices is increased to 3, 5, 7, 10, 20, 30, 40, 50 for Ds. The slice chosen randomly around the center of the object is kept the same, the additional slices are chosen randomly in the volume of the object. The propagation is started from the center slice. When a slice with given prompt is reached, this initialized prompt is used instead of the prompt generated from prediction for the propagation anew. The Dice coefficient for the different numbers of prompts is shown in Figure 5 (blue dots).

An exact time analysis of the prompt annotation process was not conducted, since the prompts have been generated automatically from the existing ground truth labels. In a real-world scenario without ground truth labels, the prompts need to be drawn manually. In order to assess the annotation time depending on the prompts used for initialization of our approaches, an annotation time estimation is calculated as:

$$(t_a \cdot c + t_s) \cdot N, \tag{5}$$

with the annotation time for one prompt per class $t_a$, the number of classes $c$, the time to scroll and navigate through the 3D volume per slice $t_s$ and the number of slices $N$. For Ds, the number of classes is 4. Under the assumption that drawing one prompt per slice takes 5 seconds per class and scrolling through the volume takes 2 seconds per slice, Equation (5) is only dependent on $N$ which we set equal to the number of initialized prompts. For Ds, the average number of slices containing all structures is maximal 269, which we use as $N$ for fully-prompted SAM. Figure 5 (red line and markers) shows the estimated annotation time for the same set of prompt numbers as used for the performance gain analysis.

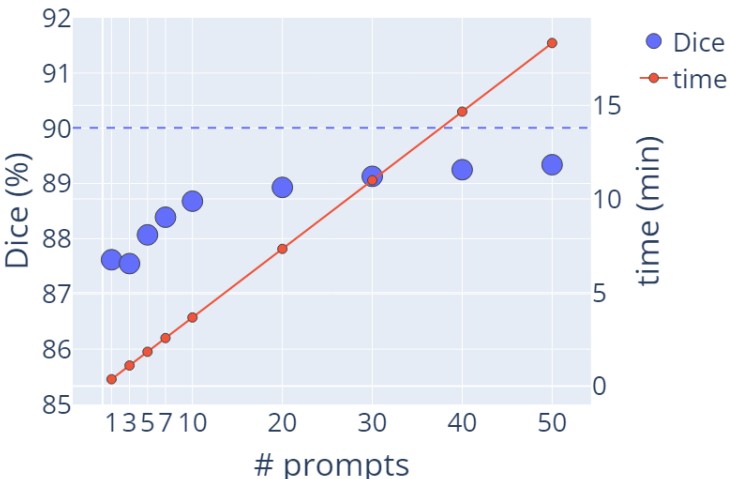

Figure 5: Analysing performance (Dice) and estimated annotation time (min) wrt different numbers of prompts given as initialization for Ds. The dotted line corresponds to the Dice value of the fully-prompted SAM.

### C.4. Performance evolution

To show the performance evolution across volumes, the performance is evaluated for different subsets of slices surrounding the slice with initial prompt. Figure 6 and Figure 7 shows the results for all four datasets and different ranges of slices. The trend is a decrease of performance since errors accumulate and are propagated towards the boundary of the object. Dataset Ds is an exception of this trend due to the discrepancy between full-bone and cortical-bone segmentation. While ground truth labels represent full-bone segmentation, predictions primarily capture cortical bone, particularly in the center of the object (see Figure 9). Increasing the number of slices towards the boundary, where predictions are filled, improves Dice scores.

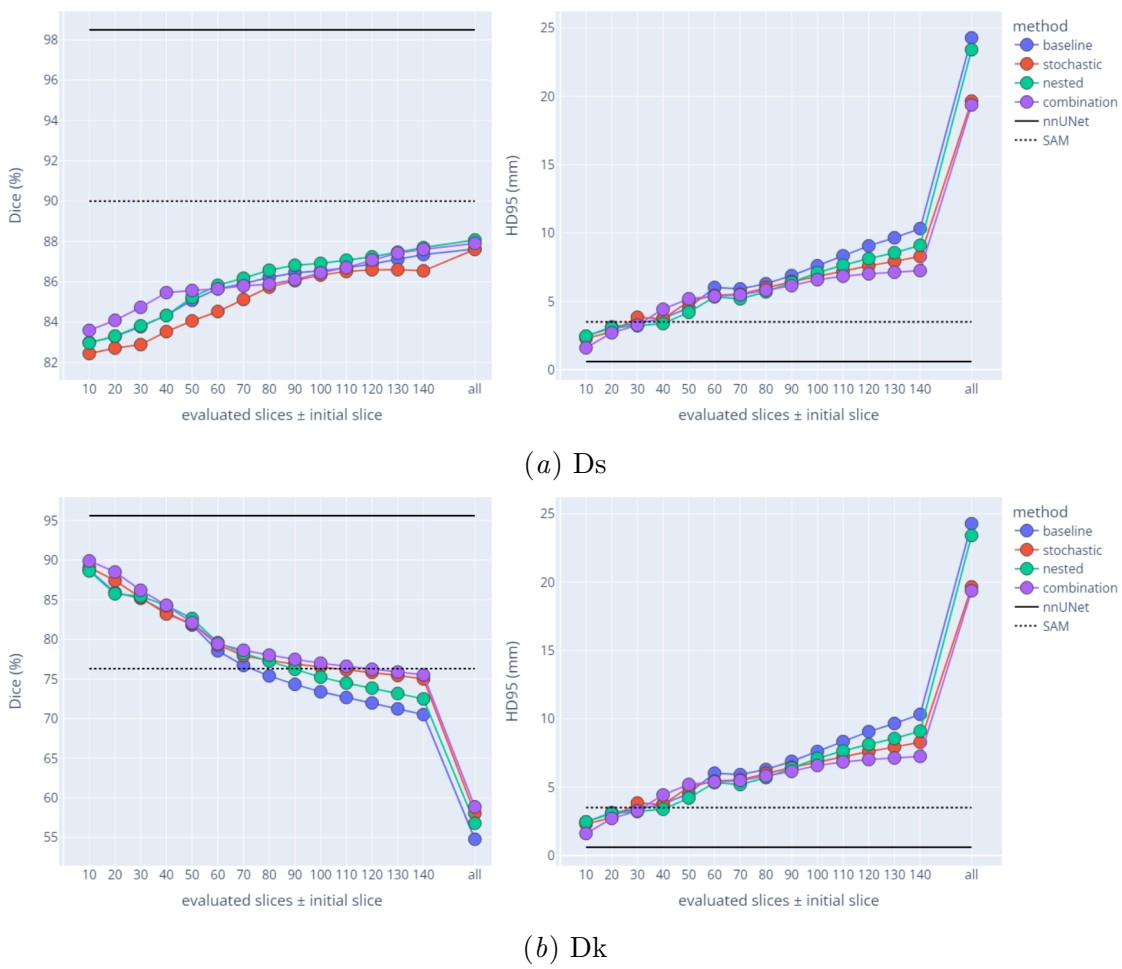

Figure 6: Analysing performance for different ranges of slices surrounding the initial slice for internal datasets Ds and Dk.

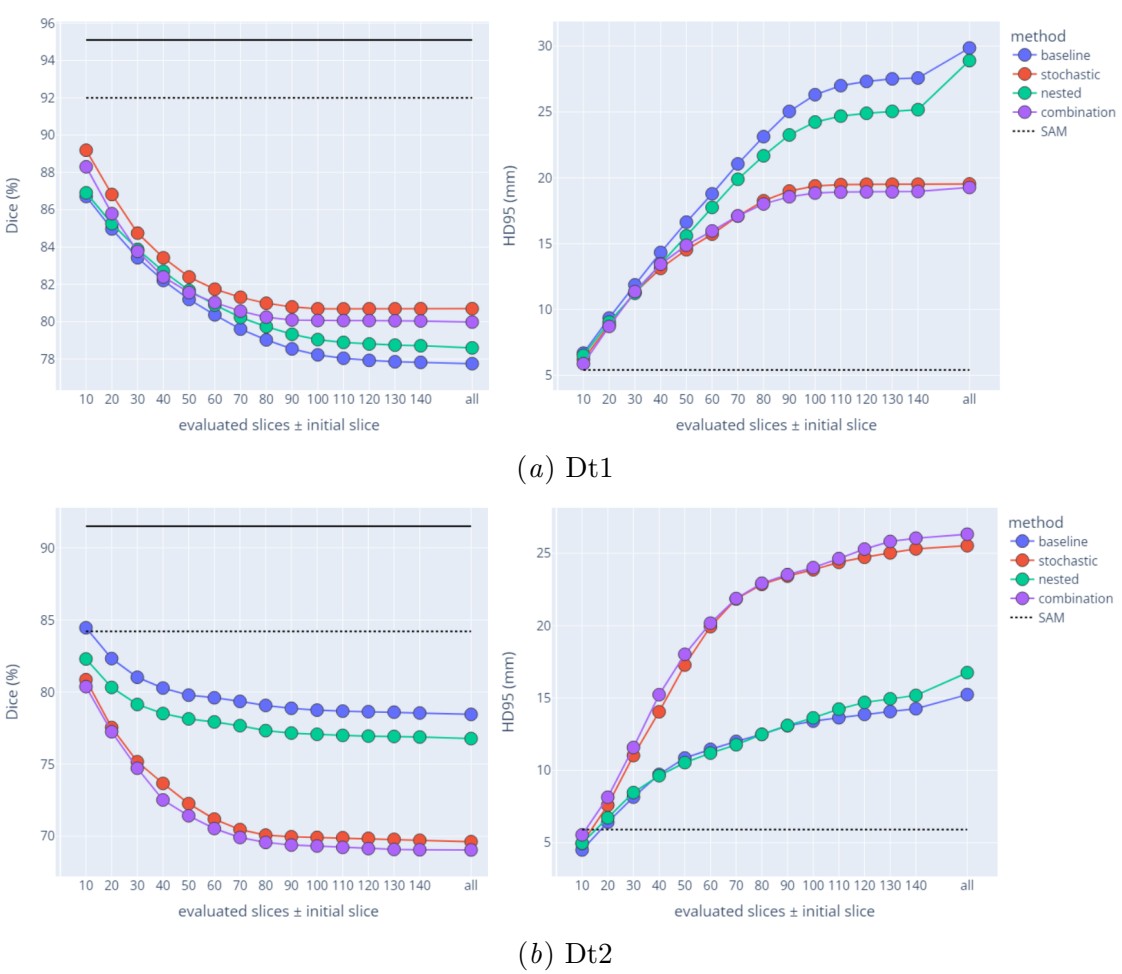

(a) Dt1

(b) Dt2

Figure 7: Analysing performance for different ranges of slices surrounding the initial slice for external datasets Dt1 and Dt2.

### C.5. Hyperparameter settings

Figure 8 shows the results for hyperparameter testing to determine the default settings for the range to randomly sample the initial slice (Figure 8($a$)), the number of iterations (Figure 8($b$)) and minimal number of models contributing to the final mask (Figure 8($c$)) in the stochastic approach.

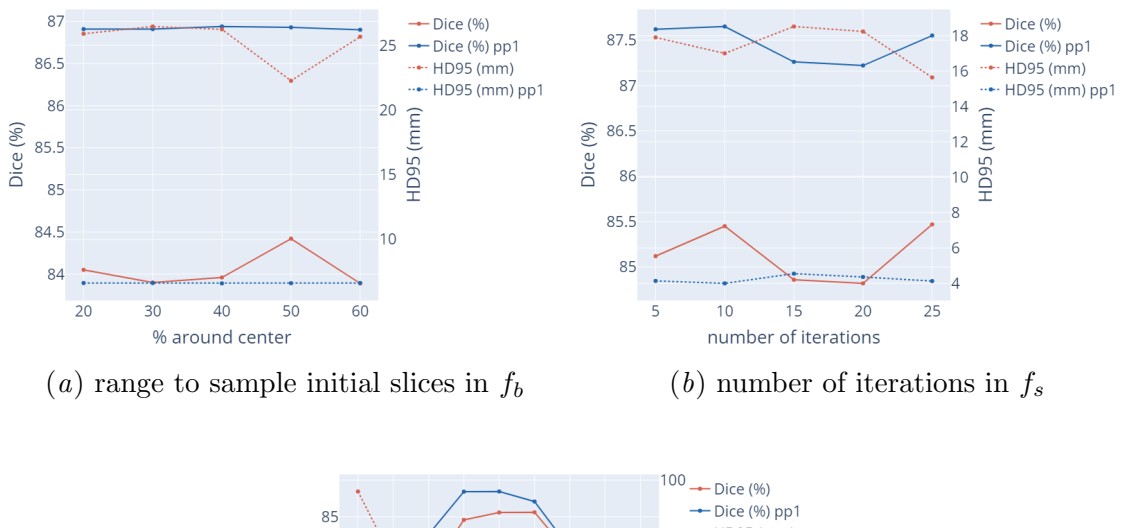

($a$) range to sample initial slices in $f_b$     ($b$) number of iterations in $f_s$

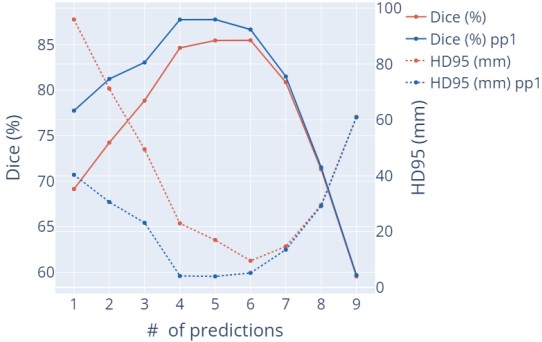

($c$) minimal number of models contributing
to the final mask in $f_s$

Figure 8: Results for hyperparameter testing on Ds

## Appendix D. Prediction examples

The following figures show examples of the predictions in different views after both post-processing steps, except when stated differently. The methods are denoted with their first letters as abbreviations: ground truth (GT), reference SAM (R), baseline (B), stochastic (S), nested (N) and combined (C) . White arrows highlight regions of interest.

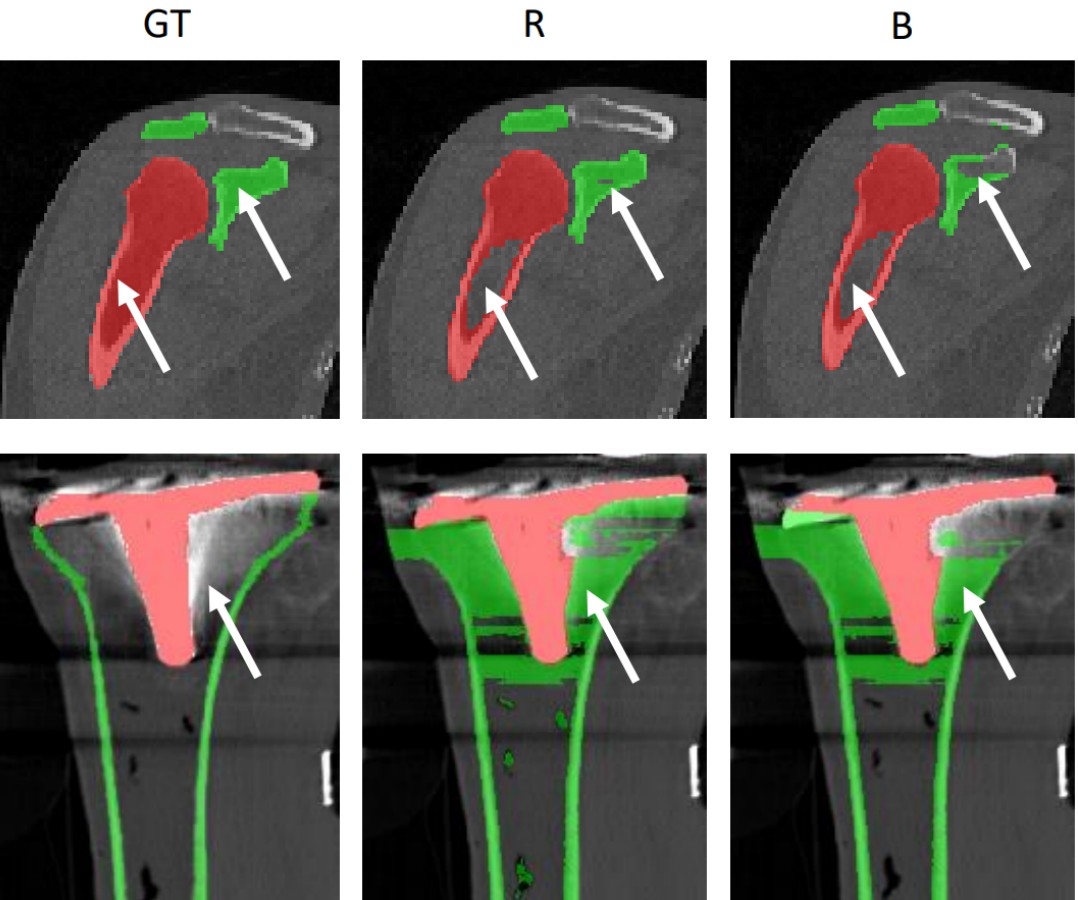

Figure 9: Cortex vs. full bone annotation protocol and predictions in coronal planes. For bone structures with visible cortex, the predictions vary between fully capturing the bone and only the bone cortex. First row: scapula (green) and humerus (red) in Ds; Second row: Tibial cortex (green) and tibial implant (red) in Dk.

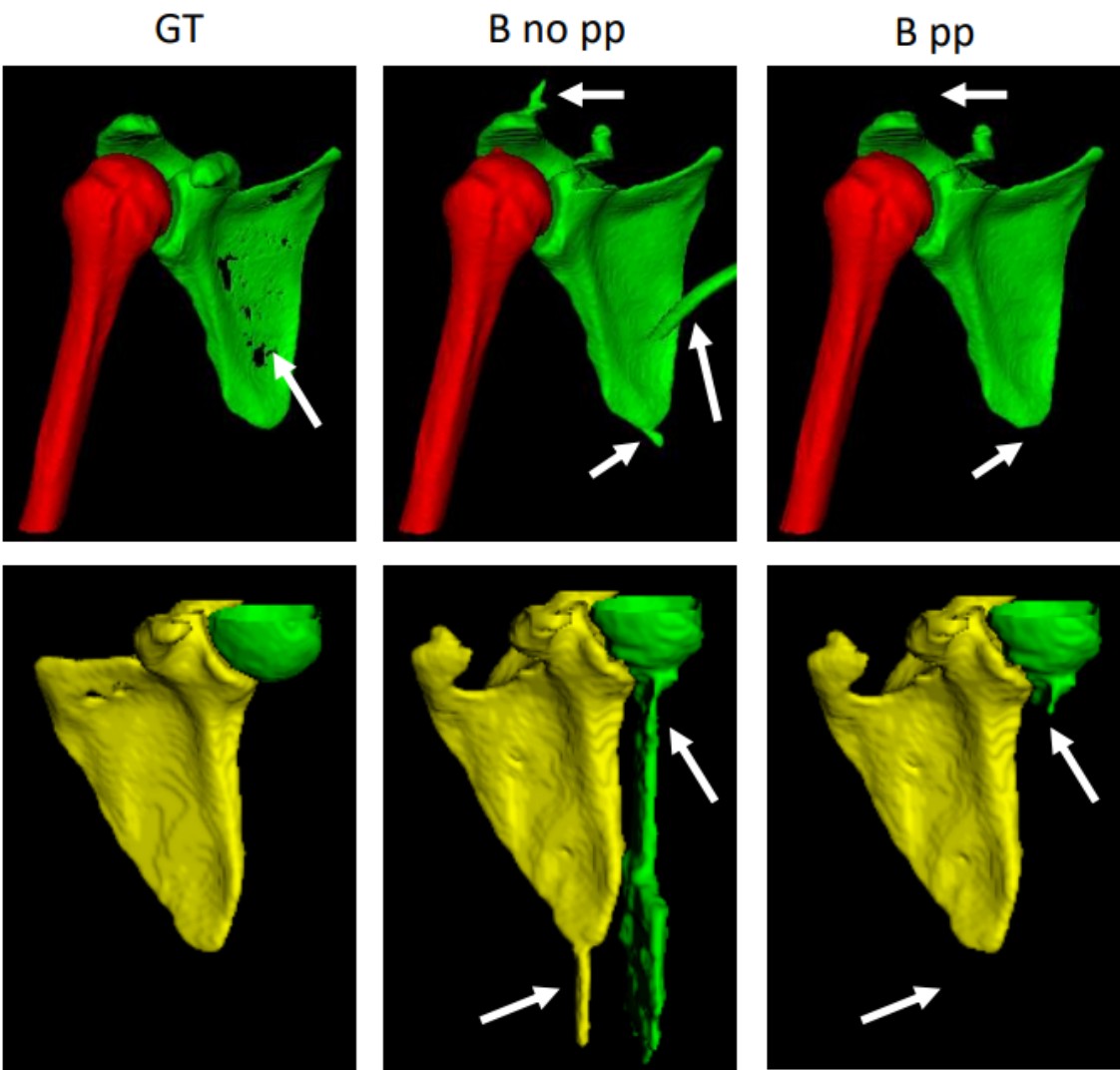

Figure 10: Affect of postprocessing shown in 3D models. First row: scapula (green) and humerus (red) in Ds; Second row: scapula (yellow) and humerus (green) in Dt1.

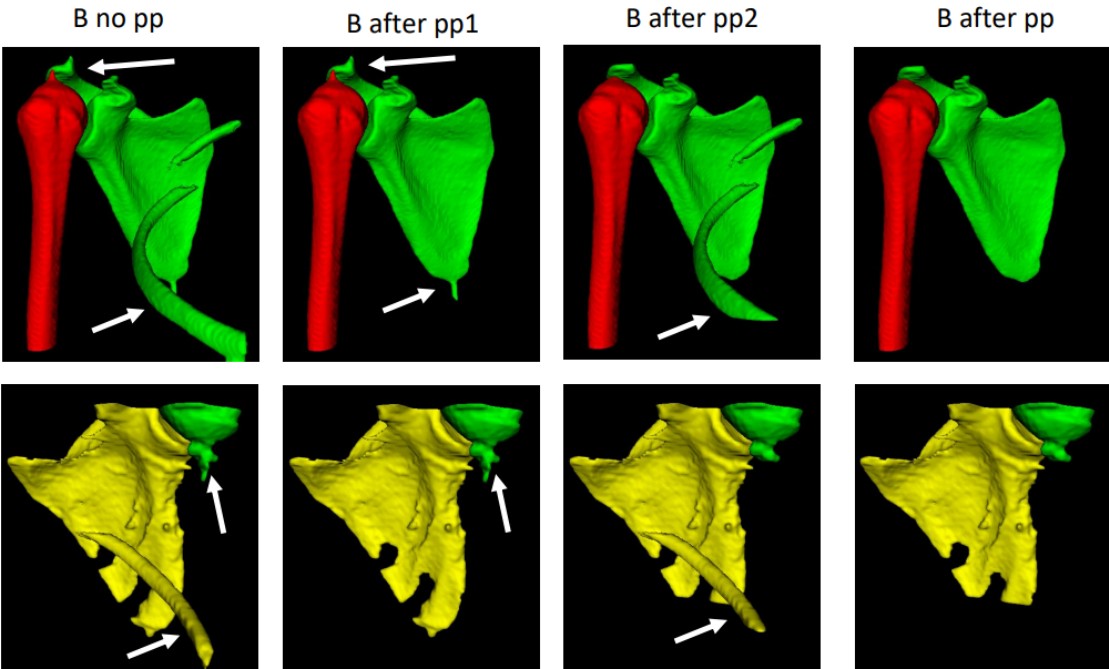

Figure 11: Affect of individual postprocessing steps shown in 3D models. First row: scapula (green) and humerus (red) in Ds; Second row: scapula (yellow) and humerus (green) in Dt1.

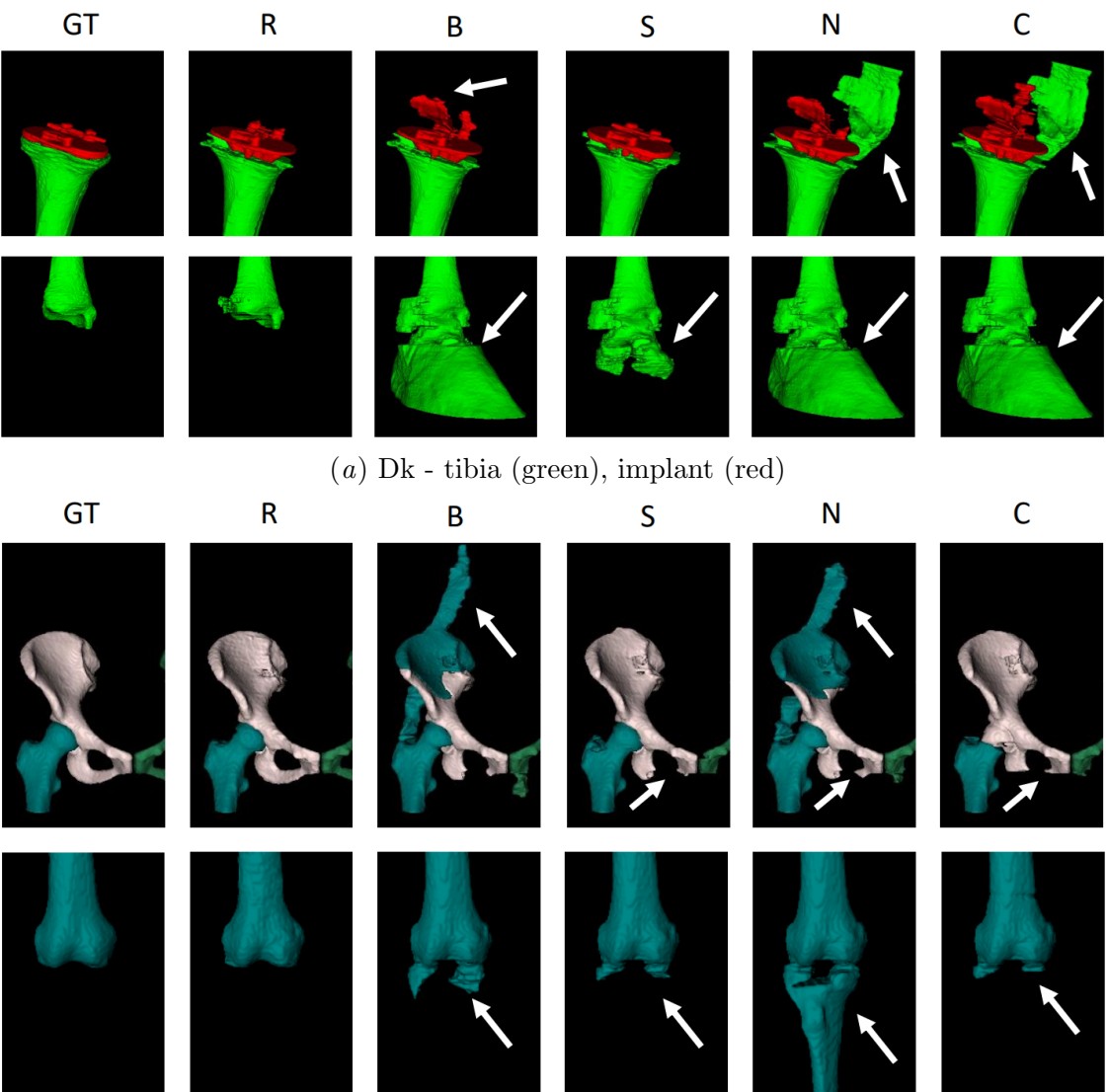

(a) Dk - tibia (green), implant (red)

(b) Dt2 - right hip (white), right femur (turquoise), left hip (green)

Figure 12: Examples of 3D models for Dk (a) and Dt2 (b). Dk shows wrong predictions of the implant plateau and overflow of the tibia to other structures, such as the ankle. Common areas of mistake for Dt2 are the ball-and-socket joint between femur and pelvis, as well as the ischium and pubis.

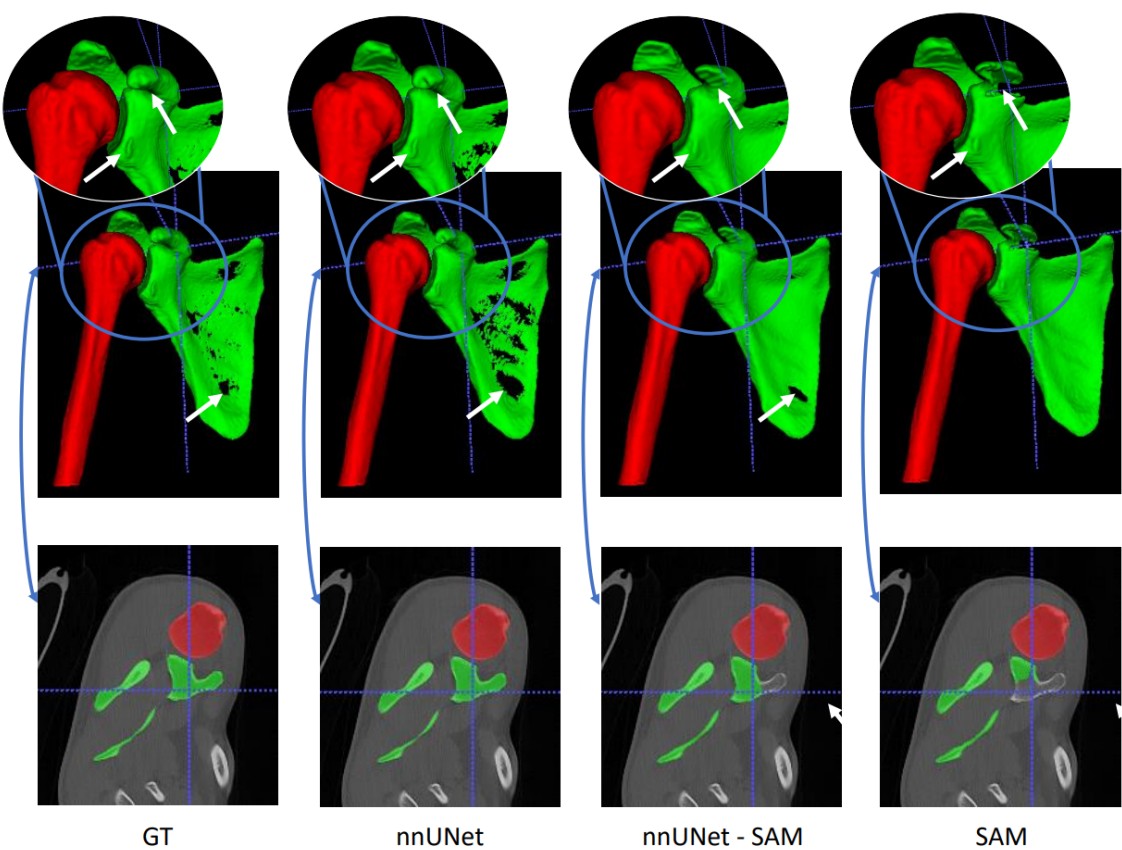

GT         nnUNet        nnUNet - SAM        SAM

Figure 13: Predictions from fully supervised models with different sets of training labels. The top row and the zoom-in show the 3D model of scapula (green) and humerus (right) with zoomed in to the humerus head. The bottom row shows the axial slice.

