# OpenReview forum: "Training-free Prompt Placement by Propagation for SAM Predictions in Bone CT Scans"
_MIDL.io/2024/Conference — MIDL 2024 Poster_

### Official Review · Reviewer_GcmQ · 2024-02-26

**Confidence:** 5
**Preliminary Rating:** 1
**Recommendation:** Poster
**Final Rating:** 1

**Summary:**

In this paper, 3 strategies for sequential box generation as input to the Segment Anything Model (SAM) are proposed for 3d CT data. 2 private and 1 public data set are analyzed. The results are compared with 3D nnUNet, and while the strategies improve blind usage of SAM, the results are far from state-of-the-art.

**Strengths:**

The method has been applied to several CT data. I'm sorry, but I don't find this to be a strong paper. I do, however, need to write at least 200 characters in the box, before I'm allowed to submit my review.

**Weaknesses:**

In general, I find the use of SAM for the problem of CT segmentation a weird choice since it is not trained for this purpose and it is known to work less than optimal as the authors also claim. It is not well argued that the choice of prompt boxes should solve this problem.

The paper contains many errors which make it difficult to read: There is a spelling error in the title "Propgation" I presume. The logical connection between the abstract's two parts before and after "In our work" is not easy to understand. The notion of "prompt placement" is not explained until late in the article, making it unclear. Reporting results on private data makes the results nonrepeatable by the reader. The term "unidirectional" above (1) is confusing since the Stochastic approach explained below is also unidirectional. The parameters for the distribution above (2) are not defined. Also in (2), I'm guessing that "with P_i ..." is to be deleted.

**Detailed Comments:**

See above

**Justification Of Final Rating:**

How to reduce the amount of human labor when training networks is an important question. From the discussions, it seems that the authors also wish to discuss the concept of Foundational Models and that this is the reason why SAM was used. However, since the paper only discusses SAM, it is not a discussion on the definition of what a Foundational Model is. And while the concept of clever usage of prompt boxing is interesting, SAM is still the wrong model for the proposed problem, analogously trying to make a car fly by showing it where to drive: Even if you drive off a cliff and fly momentarily, it still is a weird strategy for designing machines that can fly.

**Justification Of The Preliminary Rating:**

As I wrote above, I find the use of SAM for the problem of CT segmentation a weird choice since it is not trained for this purpose and it is known to work less than optimal as the authors also claim. It is not well argued that the choice of prompt boxes should solve this problem.

**Questions To Address In The Rebuttal:**

I don't find this work ready for publication, and specifically, the authors should somehow demonstrate that the choice of prompt is more important than retraining the model for the specific problem.

**Special Issue:**

No

---

> ### Author Response · Authors · 2024-03-17
> **Rebuttal for Reviewer GcmQ**
>
> We have addressed the Reviewer’s concerns by further clarifying our motivation and contribution. In addition, following the Reviewer’s suggestion, some changes to the manuscript have been made to improving clarity and readability. However, we would have appreciated a more elaborated and justified assessment in order to provide a more succinct answer to the comments.
>
> **Q1) In general, I find the use of SAM for the problem of CT segmentation a weird choice since it is not trained for this purpose and it is known to work less than optimal as the authors also claim. It is not well argued that the choice of prompt boxes should solve this problem.**
>
> A1) As the Reviewer correctly observed, SAM has been trained on natural images, not CT scans. However, SAM is introduced as segmentation FM and, by definition, FMs are capable of performing a wide variety of tasks, even those they were not explicitly trained for. The capacity for generalizability implies that it can be applied to unseen structures without requiring additional training, making it particularly valuable in domains challenged by data and annotation scarcity. Due to these attributes, the medical image segmentation community has already adopted SAM and evaluated its performance  [1,2,3,4]. The findings from previous literature studies indicate valuable contributions but significant variations in performance across different datasets and tasks. Several studies have identified specific characteristics where SAM performs better, such as clear foreground-background contrast and well-defined object borders [3,4]. These criteria are met in bone segmentation from CT scans. Considering that manual labeling of 3D CT scans is highly time consuming, specially in large structures such as bones, we believe that SAM can be effectively utilized for this task. However, we encounter two separate challenges: 1) the performance gap compared to fully supervised methods like nnUNet, and 2) the manual intervention necessary for providing prompts in the 2D architecture. The first challenge is addressed through adaptation techniques and fine-tuning, both requiring annotated datasets.
>
> Our strategies are focused on addressing the second problem, not the first. As outlined in our manuscript, our aim is not to produce new state-of-the-art results, but rather to offer an approach for utilizing prompt-based 2D FM, such as SAM, for 3D scans in an annotation-efficient way. The advantages of our strategies include being training-free and requiring only one bounding box per class per volume, which significantly reduces annotation time. However, we acknowledge that the limitations in SAM's performance cannot be overcome, given that the design of our work is centered around its usage. As the Reviewer states, our strategies improve blind usage of SAM. In our opinion, it is equally important not to blindly fine-tune SAM, but rather to prioritize informed decision-making in the fine-tuning process to optimize label and computational efficiency. Thus, we consider our results as promising because the predictions generated can be utilized in model-assisted labeling to obtain annotated datasets for fine-tuning SAM or for other fully-supervised networks. We hope that our study provides insights that can be leveraged alongside these datasets for efficient training strategies, ultimately enhancing performance.
>
> **Q2) The logical connection between the abstract's two parts before and after "In our work" is not easy to understand. The notion of "prompt placement" is not explained until late in the article, making it unclear. Reporting results on private data makes the results nonrepeatable by the reader. The term "unidirectional" above (1) is confusing since the Stochastic approach explained below is also unidirectional. The parameters for the distribution above (2) are not defined. Also in (2), I'm guessing that "with P_i ..." is to be deleted.**
>
> A2) We recognize that reporting on private data makes the results non-reproducible by the reader. In order to overcome this limitation, we included a publicly available dataset in our analysis to show results that could be reproduced by the reader. After acceptance, we also plan on making our code publicly available.
> In (2), we wanted to clarify that the prompt P_i is based on the previous segmentation mask M_i-1. We made additions to our manuscript regarding the logical connection in the abstract, the explanation of “placing a prompt”, the term “unidirectional”, and the distribution for delta (assuming that this was unclear).
>
> [1] Maciej A. Mazurowski, et al. Segment anything model for medical image analysis: An experimental study. Medical Image Analysis, 89:102918, 2023.
> [2] Dongjie Cheng, et al. SAM on medical images: A comprehensive study on three prompt modes, arXiv, 2023
> [3] Jun Ma, et al. Segment anything in medical images, arXiv, 2023
> [4] Sheng He, et al. Accuracy of segment-anything model (SAM) in medical image segmentation tasks. arXiv 2023

---

### Official Review · Reviewer_BeaE · 2024-02-28

**Confidence:** 5
**Preliminary Rating:** 2
**Final Rating:** 4

**Summary:**

-The authors introduce several automatic box-prompt generation methods for the Segment-Anything Model (SAM), given only one initial box prompt on a certain slice of a 3D volume. This would then propagate to the rest of the slices.

-The experiments were conducted on three different datasets: two in-house and one public, for bone segmentation

**Strengths:**

-The authors proposed four different training-free strategies for prompt placement by propagation to adapt the Segment-Anything Model (SAM) for bone segmentation in 3D CT scans, which saves time on annotation

**Weaknesses:**

Overall motivation (not about the performance):

-I understand the importance of training-free strategies for the field. However, I am curious about the time it would take to input additional box prompts for more slices. If it does not significantly increase the time, why not use more box prompts to enhance overall performance

-I am unclear about the next steps after segmentation is produced. I speculate that it could be utilized for pseudo-label learning. However, I question its feasibility given the unimpressive Dice scores. Additionally, considering the inherent Hounsfield Unit (HU) values make bone segmentation from CT images relatively straightforward, why not employ a pretrained model for direct segmentation.

**Detailed Comments:**

-Why not test the Segment Anything Model (SAM) solely for medical image segmentation instead of comparing it with other foundational medical models, such as 'Segment Anything in Medical Images' [1] and 'SAM MED 2D' [2]?.

-A recent work [3] aims to segment bone from MRI, I suggest author to test their methods with the pretrained SegmentAnyBone

[1] Ma, Jun, et al. "Segment anything in medical images." Nature Communications 15.1 (2024): 654.
[2] Cheng, Junlong, et al. "Sam-med2d." arXiv preprint arXiv:2308.16184 (2023).
[3] Gu, Hanxue, et al. "SegmentAnyBone: A Universal Model that Segments Any Bone at Any Location on MRI." arXiv preprint arXiv:2401.12974 (2024).

**Justification Of Final Rating:**

I was wondering if other pre-trained models for medical imaging could deliver high-quality segmentations, i.e., above 90% in terms of the Dice coefficient, or perform better than SAM. In such cases, the proposed algorithm could be developed based on these models, or we could directly use these models to obtain segmentations since generating prompts is not considered time-consuming and does not require dealing with large datasets. However, the authors have addressed my concern.

In addition, the authors provide more details on SAM Med 2D. However, the training-free strategy does not work well, and I believe it offers a valuable experimental suggestion for future studies in the field of reduced prompting. This field has not been thoroughly explored, and it is important for practical applications. This also highlights their limitations with different models.

**Justification Of The Preliminary Rating:**

The paper is well written and easy to follow. However, my biggest concern is the motivation behind the proposed submission, which influences my preliminary rating. I would like the authors to address this in their rebuttal.

**Questions To Address In The Rebuttal:**

-please see the detailed comments and weaknesses. I need the authors to address the trade-off between performance and the number of prompts, including the time required to provide them. Such an ablation study could demonstrate the contribution of the proposed work.

---

> ### Author Response · Authors · 2024-03-17
> **Rebuttal For Reviewer BeaE**
>
> We thank the reviewer for the comprehensive review and we address the questions and concerns below:
>
> **Q1) Annotation time of additional box prompts**
>
> A1) We agree with the Reviewer that more prompts are beneficial for the prediction performance, as can already been seen by the comparison between our methods and fully-prompted SAM. Based on the Reviewer’s comment, we added an analysis testing different numbers of initialized prompts for Ds (Appendix C.3.). As expected, the results show an increase in performance with increasing number of prompts.
>
> We did not perform an exact time analysis of the prompt annotation process since the prompts are generated automatically from the existing ground truth labels. However, in order to assess the annotation time depending on the prompts used for initialization of our approaches, an annotation time estimation is added in Appendix C.3. In the results, we can for example see that a gain in performance of 1.2% Dice comes with an annotation time of over 3 additional minutes based on our assumptions. It should be noted, that for other datasets, the performance gain might be higher compared to the additional annotation cost. However, this trade-off needs to be considered for downstream tasks. For example, in time constrained tasks, this time increase might not be acceptable.  The analysis as well as the insights gained from it, have been included in the manuscript.
>
> **Q2) Next steps after segmentation and feasibility.**
>
> A2) As the reviewer correctly suggested, the next step after segmentation could be pseudo-label training. The generated predictions can also be used as a starting point for model-assisted labeling process, combined with active learning strategies. In our experiment, where we trained a nnUNet on the predictions generated by our approaches, our aim was to demonstrate feasibility of one scenario, i.e. these predictions provide an approximation of the ground truth labels, enabling supervised training with significantly reduced annotation time.
>
> Aside from using the segmentations for a next step, our approaches also provide insight into the usage of SAM which can be leveraged for the selection of training samples for efficient fine-tuning. We conducted an additional evaluation of performance for various slice ranges surrounding the initial slice (Appendix C.4.). Results show that slices close to the initial prompt have more stable performance compared to boundary slices. This is due to error accumulation and propagation. An exception to that is Ds, which is discussed in C.4.. Understanding trends about performance distributions in volumes can inform slice-based correction strategies. Additionally, implementing a failure detection method atop predictions can help in identifying volumes or slices requiring human intervention (i.e., pre-select data for manual corrections). Our results indicate that some datasets or object regions have a higher need for attention to achieve satisfactory outcomes. While fine-tuning enhances performance, it demands significant data and computational resources. Thus, we believe that informed decision-making in fine-tuning is crucial for optimizing label and computational efficiency, especially in constrained domains like medical settings.
>
> The mentioning of next steps, and an additional evaluation of performance as well as the gained insights have been added to the manuscript.
>
> **Q3) Considering the inherent Hounsfield Unit (HU) values make bone segmentation from CT images relatively straightforward, why not employ a pretrained model for direct segmentation.**
>
> A3) We believe that employing a pretrained model for direct segmentation is not straightforward because of 1) differences in bone HU coming from different scanners and CT scanning protocols and 2) differing artifacts across tasks (e.g., metal artifacts in CT, as seen in Dk). Nevertheless, bone segmentation in CT scans shows some characteristics, such as a good foreground-background contrast and a distinctive object border, that makes this task relatively easy compared to other medical image segmentation tasks, e.g., lesion segmentation. Evaluation studies of SAM [1] support the notion that these data characteristics are advantageous for SAM.
>
> **Q4) Comparison to 'Segment Anything in Medical Images' [1] and 'SAM MED 2D' [2]?.**
>
> A4) We would like to clarify, that our method is not depending on the underlying 2D version of SAM and could be applied to Med-SAM [1] and SAM MED 2D [2] as well. We agree with the reviewer, an application to other FM models specialized on medical data is possible and as stated in our discussion, we would like to perform such a comparison in the future.
>
> **Q5) SegmentAnyBone**
>
> A5) Since our datasets consists of CT scans and SegmentAnyBone was trained on MRI data, we believe that the performance of the pretrained model suffers from the modality shift. To the best of our knowledge there is no equivalent of SegmentAnyBone trained on (bone) CT data.

---

> > ### Comment · Reviewer_BeaE · 2024-03-19
> >
> > Thanks for the feedback.
> >
> > Let's directly jump into the Q4 and 5.
> >
> > As many studies have shown that SAM doesn't produce good results on medical images. There are some pretrained models, with CT or without CT, publicly available for medical imaging. What if their performances are better than yours or what are the improvements between their pretrained models and your algorithm. This point makes a clear contribution of submission. I don't think the improvements based on SAM is reasonable at the current timepoint.

---

> > > ### Author Response · Authors · 2024-03-22
> > >
> > > The Reviewer is correct that the findings in previous literature studies show variations in performance across different datasets and tasks. However, as already mentioned for another Reviewer, several studies have identified specific characteristics where SAM performs better, such as foreground-background contrast and well-defined object borders [1,2]. These criteria are met in bone segmentation from CT scans. Thus, we hypothesized that SAM will show promising results on our datasets.
> > >
> > > The improvement of our method lies in the application of SAM, or other related fine-tuned versions. As stated in our manuscript, our goal is not to improve the performance of SAM, but its applicability for 3D data. By design, our strategies are limited to the underlying SAM-version. However, we believe that our training-free strategies will help improve the performance in the future, due to the potential next steps as mentioned for Q2.
> > >
> > > Nevertheless, in order to address the Reviewer’s question about a performance, we performed a preliminary comparison. The Reviewer is correct that there are pretrained models publicly available for medical imaging. On the one hand, there are pretrained models that have been trained in a fully-supervised manner on a large CT dataset using pixel-level annotations. On the other hand, Foundation Models (FM) offer generalisability and by definition, they can be applied to unseen structures without requiring additional training or annotations. Since we were not sure which of the two categories the Reviewer was referring to, we would like to discuss concrete examples from both categories. We performed preliminary evaluations for the mentioned models on our dataset Ds and we provide examples of the predictions [here]( https://www.dropbox.com/scl/fi/7xqaatmpxv4h0a6g5a0r7/MIDL_2024_examples.pdf?rlkey=g6fgx990pcxlfzawqbu6ned66&dl=0) (this link is valid until the decision deadline).
> > >
> > > **TotalSegmentator [4]**
> > >
> > > This is an example for a fully supervised segmentation network, pretrained on CT scans. Among others, it was trained on humerus and scapula segmentations. Thus, we have made preliminary experiments on our internal dataset Ds. The averaged Dice is below 50%. We believe that a difference in acquisition protocol and scanned anatomical views might be the reason for the low performance. However, this shows, that fully supervised models are specialized and cannot be transferred easily to other structures or data samples.
> > >
> > > **Med-SAM [1] and SAM Med 2D [2]**
> > >
> > > Both models were mentioned by the Reviewer as SAM version fine-tuned for medical data. We have evaluated both models in a fully-prompted manner on our dataset Ds. The average Dice is around 50% and around 75% for Med-SAM and SAM Med 2D, respectively. Both performances are far below the performance of SAM with 90%. Visual inspection of the predictions shows a non-smooth prediction with block-like structures.
> > > Although these are preliminary results of Med-SAM and SAM Med 2D on Ds, it shows that out-of-the-box application of fine-tuned SAM versions do not automatically perform better than SAM. As stated in our discussion, we believe that these need to be investigated further.
> > >
> > > **SegmentAnyBone [3]**
> > >
> > > The authors of [3] also identified the prompt-based application of SAM as a challenge and thus, introduced a fully automated mode in SegmentAnyBone. The fully automated mode is realized by hybrid prompting engineering, i.e., the prompt input during 70% of the training iterations is omitted. As already mentioned in our previous response, SegmentAnyBone was trained on MRI data and thus, we hypothesized that it does not perform well on CT data. In order to show that, we generated some predictions with the fully automated mode in a 3D and slice-by-slice manner (with and without attention mechanism), which shows very unsatisfying results (see examples). Unfortunately, the prompt-based is not further explained and we couldn't find direct support in the repository, so it was not possible at this moment to test it with our data.
> > >
> > > **Comparison on Ds**
> > > | Model            | Dice (%) | HD95 (mm) |
> > > |------------------|----------|-----------|
> > > | SAM*              | 90.0     | 2.5       |
> > > | SAM Med 2D*       | 75.3     | 9.4       |
> > > | Med-SAM*          | 51.8     | 12.1      |
> > > | TotalSegmentator | 48.8     | 186.5     |
> > > '* fully-prompted
> > >
> > > [1] Ma, Jun, et al. "Segment anything in medical images."
> > >
> > > [2] Cheng, Junlong, et al. "Sam-med2d."
> > >
> > > [3] Gu, Hanxue, et al. "SegmentAnyBone: A Universal Model that Segments Any Bone at Any Location on MRI."
> > >
> > > [4] Wassertheurer, Jakob, et al. “TotalSegmentator: Robust Segmentation of 104 Anatomic Structures in CT Images“

---

> ### Comment · Reviewer_BeaE · 2024-03-22
>
> Thanks for these. A quick question: what's the performance of your algorithm based on pretrained sam med 2D.

---

> > ### Author Response · Authors · 2024-03-22
> >
> > Based on the Reviewer’s question, we have evaluated our strategies based on the pretrained SAM Med 2D on Ds. The results after postprocessing can be found in the table below with the comparison to SAM performances for more convenience.
> >
> > Examples for all four strategies can be found [here](https://www.dropbox.com/scl/fi/6dhw3oseewgm5voavuce5/MIDL_2024_examples02.pdf?rlkey=dfu49g8moir8pfsrzclh3aq8j&dl=0) (this link will be valid until the decision deadline).
> >
> > | Method                    | Dice (%) | HD95 (mm) |
> > |---------------------------|----------|-----------|
> > | SAM fully-prompted        | 90.0     | 2.5       |
> > | SAM baseline              | 87.6     | 4.2       |
> > | SAM stochastic            | 87.6     | 4.0       |
> > | SAM nested                | 88.1     | 4.1       |
> > | SAM combined              | 87.9     | 4.5       |
> > |                           |          |           |
> > | SAM Med 2D fully-prompted | 75.3     | 9.4       |
> > | SAM Med 2D baseline       | 43.2     | 63.9      |
> > | SAM Med 2D stochastic     | 46.0     | 60.6      |
> > | SAM Med 2D nested         | 46.3     | 56.1      |
> > | SAM Med 2D combined       | 44.5     | 61.2      |

---

> > > ### Comment · Reviewer_BeaE · 2024-03-22
> > >
> > > Thank you for addressing my concerns. It appears that your algorithm does not work with different models (the inconsistent performance), which might be a limitation worth mentioning in the paper, even if only in a sentence. The area of training-free or reduced prompting has not been well explored, and I believe it presents a valuable experimental suggestion for future research. Good luck!

---

> > > > ### Author Response · Authors · 2024-03-25
> > > >
> > > > Our latest analysis shows that the performance of our strategies indeed depends on the underlying model, i.e., SAM and SAM Med 2D, which should be further investigated. Considering the observed drop in performance with pretrained models, such as SAM Med 2D, the performance of SAM Med2D with our strategies is consequently affected. We believe that this observation is indeed worth mentioning. If we are still allowed to make a change to our manuscript, we would mention this observation in our discussion.
> > > >
> > > > We thank the Reviewer for the discussion which helped us making this observation.

---

### Official Review · Reviewer_HjpC · 2024-02-29

**Confidence:** 4
**Preliminary Rating:** 4
**Recommendation:** Poster
**Final Rating:** 5

**Summary:**

This paper introduces a training-free approach for extending 2D SAM to 3D segmentation using prompt propagation, reducing labeling efforts from pixel-level annotations to a single initial box prompt. The exploration of four box prompt placement strategies aims to optimize segmentation performance. The proposed method demonstrates competitive results against the nnUNet benchmark and fully-prompted SAM, underscoring the potential of prompt-based methods in 3D medical segmentation.

**Strengths:**

1. The paper presents a clear and innovative solution for 3D segmentation using a 2D SAM model, leveraging a single box prompt.
2. It significantly reduces annotation requirements through four strategic approaches to box prompt propagation.
3. The method showcases promising segmentation outcomes without necessitating a training phase.

**Weaknesses:**

1. The distinctions among the three prompt propagation strategies appear minimal, lacking a definitive conclusion on their comparative effectiveness.
2. A noticeable performance gap persists when compared to nnUNet and fully-prompted SAM, indicating room for improvement.

**Detailed Comments:**

1. The paper should provide insights or discussions on instances where pp2 underperforms pp1.
2. Clarification on the statistical significance (p-values) between different methods in Table 1 would enhance the credibility of the results.

**Justification Of Final Rating:**

The author addressed my concerns with detailed illustrations. The paper provides experimental results for future work on reducing the annotation workload with SAM-based methods, leading me to increase the score in the final judgment.

**Justification Of The Preliminary Rating:**

While there is a performance disparity with nnUNet and fully-prompted SAM, the concept of training-free 3D segmentation using a single box prompt remains compelling as a methodology paper. With further illustration and discussion of the four box prompt placement strategies, the paper can still provide valuable insights for the community as a validation study.

**Questions To Address In The Rebuttal:**

While there's a performance disparity with nnUNet and fully-prompted SAM, the concept of training-free 3D segmentation using a single box prompt remains compelling. Addressing the questions raised about the weaknesses and detailed comments, along with a more thorough discussion and theoretical illustration for the box placement strategies, could substantially improve the manuscript's contribution as a methodological and validation study.

**Special Issue:**

No

---

> ### Author Response · Authors · 2024-03-17
> **Rebuttal for Reviewer HjpC**
>
> We thank the Reviewer for the helpful comments and comprehensive review.
>
> **Q1) The distinctions among the three prompt propagation strategies appear minimal, lacking a definitive conclusion on their comparative effectiveness. Clarification on the statistical significance (p-values) between different methods in Table 1 would enhance the credibility of the results.**
>
> A1) We agree that the differences between the prompt propagation strategies could be highlighted. For that, we have included Figure 1 in the manuscript illustrating each strategy. Additionally, we have performed an analysis on statistical differences between the proposed strategies using bootstrapping. After Bonferroni correction, it is shown that no statistical differences could be demonstrated. It is important to note that the number of samples is very small in each of the datasets.
> However, qualitative analysis of the results showed noticeable differences of their effectiveness. Specifically, the baseline and nested approach suffer in some cases from over-segmentation. The reason for that is the error propagation, especially at the top or bottom of an object. The bounding box is not disappearing and includes mainly background (see Figure 10). In contrast, the stochastic approach is more conservative, suffering from under-segmentation in some cases (see Figure 10). The reason is that the required agreement between multiple predictions can lead to vanishing boxes if this agreement is missing.
>
> The statistical analysis can be found in Appendix C.1. The general analysis of the approaches and discussion of visual inspection is discussed in Section 6 (3rd paragraph).
>
> **Q2) A noticeable performance gap persists when compared to nnUNet and fully-prompted SAM, indicating room for improvement.**
>
> A2) The Reviewer is correct that a performance gap still remains between nnUNet and FM as previously reported in literature [1]. The goal of our work was 1) to shed light on the applicability of SAM in medical imaging, i.e., specifically the need of a 3D approach, and 2) to provide efficient, training free strategies to facilitate this application. Although we managed to show that it is possible to have an approach to segment 3D scans with almost negligibly annotation time, in new experiments (see Appendix C.3.), we saw that the performance gap to fully-prompted SAM could be reduced by increasing the number of needed prompts.
>
> Please note that our study did not initially aim at closing the performance gap between SAM and nnUNet, but we believe it opens the way for more efficient strategies for training segmentation approaches. As also commented by another Reviewer, our predictions can be used for various follow-up steps, among others pseudo-label learning or model-assisted labeling. As shown in an experiment in the discussion section, the obtained training-free segmentations can be used as training samples for fully-supervised approaches and the a performance of 90% Dice has been achieved. On the one hand, this helps identifying labels for fine-tuning FM in a more efficient way than just random selection as currently done. For example, in our dataset Dk we would choose to annotate the tibia bone, as this is a label that SAM doesn’t perform well on. Additionally, it gives insight which datasets or tasks even benefit from fine-tuning and which already show sufficient or stable performance. We added a discussion of next steps in our manuscript.
>
> [1] Cheng, Dongjie et al., SAM on medical images: A comprehensive study on three prompt modes, 2023
>
> **Q3) The paper should provide insights or discussions on instances where pp2 underperforms pp1.**
>
> A3) We agree with the Reviewer that our representation of the results may have caused some misunderstanding regarding the application of the postprocessing steps. To clarify the result representation, Table 1 has been adapted to only show results without postprocessing and with both postprocessing steps (pp1+pp2). Results from only applying pp1 or pp2 are included in the Appendix C.2. as ablation study. In addition, an example comparing the predictions without pp, with pp2, with pp1+pp2 (Figure 8 ) and an example without pp, after pp1, after pp2, and after pp (pp1+pp2) (Figure 9) are added to the appendix. From the results and the example, it can be seen that only applying pp2 underperforms pp1. The reason is that corrections made in pp1 are larger than for pp2. In pp1, disconnected structures, e.g., the rib in scapula predictions in Ds, are removed. These are commonly larger structures. In contrast, the corrections in pp2 are mainly made in a few slices at the top and bottom of the object, with fewer pixels affected. Following the Reviewer’s suggestions, these insights are added to the discussion.

---

### Author Response · Authors · 2024-03-17
**General Comment**

We thank the Reviewers for their valuable feedback, which helped us improve the paper. We have carefully addressed each of their insights and suggestions, enhancing the clarity and exposition of the contribution of our work. We would have appreciated, though, a more elaborated and justified feedback by Reviewer GcmQ in order to provide this Reviewer a more succinct answer. The updated manuscript includes now insights about the differences between the prompt propagation strategies, as well as between postpocressing steps, with new experiments and qualitative analysis. Additionally, we have extended the quantitative analysis of the trade-off between annotation time and performance gains with additional prompts. Finally, we have expanded upon the significance and applicability of the results to enhance labeling efficiency, providing quantitative insights of its potential. We believe that these improvements have considerably enhanced the paper, adequately addressing all concerns brought up by the reviewers.

---

### Comment · Area_Chair_WSzZ · 2024-03-19
**The discussion period begins**

Dear reviewers and authors,

Thank you for your contribution to MIDL24. The discussion period begins! I encourage all reviewers and authors to participate in the discussion to address questions and clarify uncertainties.

Thank you!

---

### Meta-Review · Area_Chair_WSzZ · 2024-04-03

**Recommendation:** Accept (Poster)
**Confidence:** 5

**Metareview:**

The proposed work presented several prompt strategies to leverage the 2D SAM model for 3D bone segmentation. In the experiment, the authors evaluated these strategies and provided comparisons with other methods.

Reviewers HjpC and BeaE agreed that the proposed work facilitates the generation of pseudo labels, thereby reducing the annotation time. However, two reviewers (GcmQ and BeaE) also questioned the idea of using SAM for CT segmentation, noting that SAM was not originally designed for medical imaging. I also agree with the reviewer BeaE that an existing pretrained out-of-box tool,  such as TotalSegmentator (TS), can segment bone structures as discussed in this paper. I am particularly curious why the TS only achieved the DSC of 0.5 on the internal Ds dataset. It might be due to the domain shift issues, as explained by the authors.

Nevertheless, as mentioned in the rebuttal, this paper aims to demonstrate how SAM can be used for medical image segmentation through different prompts strategies. The proposed paper empirically and effectively showcases this concept. As pointed out by one reviewer, applying vision foundation models in medical imaging is still new. This manuscript provides valuable insights for other researchers to explore this emerging field.

---

### Decision · Program_Chairs · 2024-04-05

Accept (Poster)